# Lifestyle Modification for Enhancing Autonomic Cardiac Regulation in Children: The Role of Exercise

**DOI:** 10.3390/children6110127

**Published:** 2019-11-16

**Authors:** Kathryn E Speer, Nenad Naumovski, Stuart Semple, Andrew J McKune

**Affiliations:** 1Faculty of Health, Discipline of Sport and Exercise Science, University of Canberra, Canberra (ACT) 2617, Australia; stuart.semple@canberra.edu.au (S.S.); andrew.mckune@canberra.edu.au (A.J.M.); 2Research Institute for Sport and Exercise Science, University of Canberra, Canberra (ACT) 2617, Australia; 3Faculty of Health, University of Canberra, Canberra (ACT) 2617, Australia; nenad.naumovski@canberra.edu.au; 4Discipline of Biokinetics, Exercise and Leisure Sciences, School of Health Sciences, University of KwaZulu-Natal, Durban (KwaZulu-Natal) 4041, South Africa

**Keywords:** physical activity, exercise, autonomic nervous system, heart rate variability, children

## Abstract

Decreased physical activity (PA) is a global concern contributing to the rise in cardiometabolic diseases. One potential mechanism linking insufficient PA and poor health is dysregulated autonomic nervous system (ANS) activity. This relationship is established in adults and PA recommendations, with specific exercise prescription guidelines, have been proposed to overcome this societal health burden. However, research on the benefits and underlying mechanisms of exercise on ANS activity in children <18 years old is limited. This review aimed to describe the optimal exercise “dose” and potential mechanisms of action that exercise may pose on enhancing child ANS activity, represented by heart rate variability (HRV). PubMed, Web of Science and Google Scholar were searched for articles examining the influence of exercise on child HRV. Various exercise duration and frequency combinations appear to improve HRV indices, primarily those representing parasympathetic influence. Furthermore, both aerobic and resistance training benefit HRV through potentially different mechanisms with intensity proposed to be important for exercise prescription. Findings indicate that exercise is a crucial lifestyle modification with protective and therapeutic effects on cardiometabolic health associated with improvements in child ANS activity. Exercise programming must consider the various components including mode, intensity and population characteristics to optimize ANS health.

## 1. Introduction

Physical activity (PA) is frequently reported to be associated with protective, moderating and therapeutic effects on overall health and life expectancy [1]. However, 60% of the global population is estimated to be physically inactive with inadequate participation contributing to 6–10% of the non-communicable disease burden and 9% of early mortality [2,3]. According to several international PA report cards, there is proposed to be an “*inactivity crisis*” amongst both adults and children alike [4,5,6,7]. This crisis is mainly characterized by decreasing PA participation with approximately 80% of Asian, Australian and American children not meeting the recommended exercise guidelines and increased participation in sedentary behaviors [4,5,6]. Decreased engagement and participation in PA or exercise is a concerning modifiable lifestyle factor in children given the crucial role it plays in augmenting risk of physical and psychosocial disorders (i.e., cardiovascular disease, obesity, diabetes mellitus (DM), major depressive disorder, dementia, socializing issues) [8,9,10]. 

An underlying mechanism for the association between inadequate PA or exercise participation and negative health outcomes may be a dysregulated cardiac autonomic nervous system (ANS), as measured by a marker of ANS cardiac regulation, heart rate variability (HRV) [11]. Typically, at rest, the parasympathetic nervous system (PNS) has a greater contribution to total ANS activity whilst the sympathetic nervous system (SNS) is less active [12]. Whilst resting HRV measures are important, its reactivity may provide better clinical insight than resting HRV indices into ANS cardiac regulation [11,12]. Specifically, child HRV reactivity to stress (e.g., exercise) indicates ANS flexibility such that a slower transition from SNS dominance/vagal withdrawal to PNS dominance/vagal activation after termination of the stressor is indicative of poorer health outcomes [12].

One lifestyle modification for enhancing ANS regulation in children is proposed to be via the increase in appropriate exposure to exercise, which has been shown to benefit overall health and wellbeing [13]. Moreover, low PA participation appears to be associated with decreased PNS activity/cardiac vagal tone and increased prevalence of diseases associated with cardiac autonomic neuropathy (i.e., cardiovascular disease, DM, hypertension) [14,15,16]. Although previous findings indicate a positive relationship between exercise and HRV, optimal and safe training loads for enhancing ANS cardiac regulation should be considered as over-training or insufficient recovery may potentially lead to dysregulated ANS activity [17,18]. Additionally, for the HRV-derived PNS activity, lower values suggest less ANS flexibility (i.e., regulation/adaptability) and a greater risk of negative health outcomes [19]. Various exercise interventions with different training types and loads appear to impact ANS cardiac regulation in children. In this article we examine the current literature in the form of a narrative review investigating the potential protective and therapeutic impacts of exercise for enhancing ANS cardiac regulation in children. As exercise intervention and participation is a rather complex process that involves several components, this review has been divided into broad categories related to the exercise interventions: exercise dose, mode, short vs. long-term interventions, and exercise in clinical and sub-clinical populations. Furthermore, we discuss the potential mechanisms of action and exercise recommendations for children in order to enhance their ANS cardiac regulation.

## 2. Methods

A literature search was performed from inception to September 2019 for articles examining the influence of exercise (i.e., dose, mode, intervention duration) on HRV in children under 18 years old. Articles reporting human studies and published in the English language were retrieved from the following three electronic databases: PubMed, Web of Science, and Google Scholar. Search terms included for article retrieval were “*autonomic nervous system*” OR “*heart rate variability*”, AND “*exercise*” OR “*physical activity*” OR “*aerobic exercise/training*” OR “*resistance exercise/training*” OR “*exercise intensity*”, AND “*children*”. A hand search of the reference lists from the retrieved articles was also undertaken to ensure all relevant articles had been captured by the search strategy. Two authors independently reviewed the retrieved articles to determine the final selections. The literature search retrieved 1989 articles, of which 56 full texts were analyzed with 13 articles accepted for inclusion in this review (Appendix A).

## 3. Exercise Dose

Exercise dose describes the amount of work completed over a period of time and incorporates the duration and intensity (i.e., degree of effort exerted during exercise) of each training session as well as the frequency (i.e., number) of sessions per week. Global guidelines recommend that children aged 5–17 years old should engage in at least 60 min of moderate to vigorous PA daily with higher training loads and intensities imparting greater health benefits [20]. Regarding childhood exercise interventions for enhancing ANS cardiac regulation, frequency of sessions ranged from 3 days per week to daily sessions with individual session durations ranging from 20 min up to 2 h [14,18,21,22,23,24,25,26,27,28]. Intensity of sessions were typically performed at a moderate to vigorous percentage of heart rate reserve (40–85%) [29].

Whether the frequency of training sessions was a minimum of 3 days/week or daily training sessions, findings consistently displayed positive effects on child ANS cardiac regulation (i.e., increased PNS parameters and total HRV) [14,21,22,23,24,25,26,27,28]. Similarly, training sessions from 20 min up to 2 h positively impacted child cardiac ANS regulation [14,21,22,23,24,25,26,27,28]. However, the intensity of exercise sessions (i.e., moderate, vigorous, maximal, and supramaximal) may have an essential influence on child ANS regulation and, thus, PA guidelines that only give recommendations based on total daily volume are inadequate. [17,18,30]. 

In a study by Gamelin et al., after a 7-week intervention of maximal and supramaximal aerobic exercise training of a relatively low training load (i.e., three, 30-min sessions/week), there were no significant differences in HRV (and no improvements) in the exercise intervention group as compared with the control group [18]. The authors proposed that the 7-week intervention may have been too short to improve HRV, although shorter, aerobic exercise investigations involving similar session duration and sessions/week demonstrated enhanced cardiac vagal regulation in their participants [14]. A notable difference in the exercise dose prescribed by Gamelin et al. compared with other studies was the maximal and supramaximal exercise intensity. Indeed, the lack of difference/improvement of HRV in the exercise intervention group may have reflected exercise- induced fatigue and/or an over-reaching effect on the ANS, masking the potential HRV improvements [17,18]. Due to the high intensity of work, participant HRV recovery may not have been evident due to temporary metabolic/physiological disturbances (i.e., increased blood pressure, cardiac output and muscle lactate, depletion of muscle glycogen, hypoxia) [31,32]. 

Currently, research investigating maximal and supramaximal exercise intensity in children is limited. However, such research may explain potential mechanisms underlying the importance of exercise intensity and how that affects HRV reactivity by including follow-up HRV measurements (i.e., hours, days and/or weeks after completion of the intervention). Indeed, intense aerobic exercise may induce a temporary state of hypoglycemia, leading to reduced brain glycogen stores and increased lactate crossing the blood-brain barrier [33]. Moreover, studies in animal models have indicated that increased concentrations (i.e., above normal, resting levels) of brain lactate and exercised-induced neurotrophins remain high from 50 min to two weeks following acute exercise [33,34,35]. Given that neurons in the ANS have been proposed to use lactate rather than glucose as a primary energy source, increased concentrations of brain lactate may sustain activation of the SNS [33,34]. Therefore, maximal and supramaximal exercise in children may augment extracellular blood- brain lactate levels, leading to sustained SNS activation and a pro-inflammatory state persisting after exercise cessation [33,36]. This pro-inflammatory state may delay autonomic recovery from exercise via transient stress placed on various neuroendocrine and baroreflex-modulated systems [37]. Alternatively, follow-up HRV measurements may indicate that maximal and supramaximal exercise levels exceed an intensity threshold for optimal ANS cardiac regulation in children [17]. Furthermore, whilst most studies suggest moderate to vigorous exercise levels to be most beneficial for remodeling and/or improving ANS activity, it would be beneficial for future studies to investigate whether low exercise dose interventions (i.e., 20–30-min low intensity sessions once or twice per week) contribute to positive ANS adaptations [14].

## 4. Exercise Mode

Traditional exercise modes outline the type of exercise incorporated into an intervention or training regime. The type of exercise(s) typically consists of various movement patterns/activities for improving specific bodily systems [38]. For the purpose of this review, the bodily systems targeted for enhancing child cardiac ANS regulation were cardiorespiratory (i.e., aerobic training) and muscular/bone (i.e., resistance training and flexibility/balance training). It appears that aerobic training (i.e., walking, running, cycling, dancing, skipping) has been the most common exercise intervention investigated in children. This may be due to ease of implementation such that aerobic training is usually more habitual, automatic, and requires less attention, quick decision-making, balance and coordination at a high metabolic cost to the individual [38]. Several studies within the last decade have demonstrated improvements in ANS cardiac regulation (i.e., higher PNS and lower SNS indices at rest) in children ranging from 10–14 years old [14,28,39]. It is possible that aerobic training imposes physical/structural changes on the cardiorespiratory system (i.e., heart and lungs), which may improve oxygen uptake and cardiac output efficiency [40]. Aerobic training induced cardiorespiratory modifications might remodel ANS cardiac regulation by increasing exposure to high metabolic demands which the ANS must meet via baroreflex mediation and recover from through vagal modulation [39,40]. Furthermore, these effects seemed to be independent of diet as demonstrated by a 4-month intervention comparing ANS activity of obese children given a hypocaloric diet combined with an aerobic exercise training program with matched children only given the hypocaloric diet [28]. Prado and colleagues (2010) observed improved cardiac ANS regulation in the combined diet and exercise training group as compared with diet only group [28]. Thus, findings may indicate that aerobic exercise remodels the cardiorespiratory system, resulting in improved sympathovagal balance and ANS cardiac regulation from high metabolic demands and baroreflex conditioning that weight reduction alone cannot impart [25,28,38]. 

Whilst aerobic exercise has been suggested to benefit cardiac ANS regulation in children, resistance training should also be investigated. Although it is harder to implement as an intervention due to increased neuromuscular undertaking, resistance training may also enhance ANS cardiac regulation in children. In a study conducted by Farinatti et al., resistance training (targeting both upper and lower body strength movements) demonstrated ANS remodeling in obese children by increasing PNS activity/vagal modulation, such that after the intervention, differences between PNS parameters were no longer evident between the obese and healthy children [27]. Similarly, a study by Patil et al. implemented a yoga-based intervention on sub-junior cyclists and demonstrated significant improvements (*p* <0.05) on ANS cardiac regulation in the group that received the yoga intervention compared with the group that did not [24]. Considering strength-based resistance training, it is possible that the intervention counteracted ANS dysregulation in obese children by stimulating the exercise pressor reflex via increased muscle stimulation [27,41]. In conjunction with SNS activation during strength-training, the working muscle would have stimulated baroreceptors to modulate blood pressure, vascular resistance, and cardiac output, resulting in improved PNS activity after the intervention ceased [27,41]. Alternatively, resistance training could enhance ANS regulation through the relationship between lactate, brain-derived neurotropic factor (BDNF) and neurogenesis. The moderate–vigorous circuit styled resistance training intervention (as opposed to an aerobic-based intervention) administered by Farinatti et al. may have generated a larger lactate flux across the blood–brain barrier from increased blood flow due to greater muscle mass activation [42,43]. As the preferred fuel source over glucose for neurons, lactate influx into the central nervous system in response to resistance-based exercise enhances secretion of BDNF which may aid in ANS and vascular enhancement [33,42]. Favorable influence of BDNF on ANS cardiac regulation may be attributed to the potential direct effects of BDNF for optimizing cell energy metabolism (i.e., insulin sensitivity, brown fat formation, cardiovascular strength/resilience), synaptic plasticity (neuronal growth, repair, and mitochondrial efficiency) and neurogenesis within various neuroendocrine pathways including the cholinergic neurons within the ANS [35,44]. On the other hand, enhancement of ANS cardiac regulation in children through yoga practice may be attributed to stress reduction mechanisms rather than as a mode of exercise. The authors reported that yoga may have acted as a coping mechanism for stress such that reductions in cortisol and various catecholamine levels resulted in decreased SNS and increased PNS contribution of total HRV [24]. Importantly, the authors did not incorporate other markers of stress into their study so reductions in stress hormone levels is speculative. Interestingly, the sub-junior cyclists that did not receive the yoga intervention demonstrated significantly decreased indices of PNS activity after the training period which may illustrate an over-training effect that regular yoga practice may help to counteract [17].

## 5. Intervention Duration

Studies investigating the relationship between exercise and child ANS cardiac regulation appear to conduct interventions ranging from 2 weeks to 12 months in children as young as 6 years old. Most studies seem to demonstrate that exercise interventions enhanced ANS cardiac regulation (at post- intervention measurements) in children regardless of the duration [14,21,22,23,24,25,26,39]. However, findings from a 7-week aerobic exercise intervention conducted by Gamelin et al. did not support this notion [18]. Instead, the authors found no significant differences in HRV between the exercise intervention and control group, attributing their lack of significant group difference to too brief an intervention period to modify ANS regulation [18]. A study by Bond et al. exhibited PNS enhancement in adolescents after two weeks of aerobic training, however, most of the improvements dissipated at the 3-day post-intervention follow-up [39]. Of note, is that many studies examining the effect of exercise on the enhancement of ANS cardiac regulation enhancement in children only compare pre- and post-intervention HRV measurements. Future studies should implement follow- up measurements, which would potentially identify evidence-based exercise interventions for enhancing ANS cardiac regulation in children. Furthermore, a study by Gutin et al. demonstrated that ANS improvements following a longer term exercise intervention of 4 months duration were negated when HRV was measured 4 months after the intervention ceased [22]. Nonetheless, this finding highlights the importance of continuous exercise participation to enhance and maintain enhanced ANS cardiac regulation in children.

## 6. Exercise in Pediatric Clinical Populations

Exercise may also have therapeutic effects on chronic conditions that impact ANS cardiac regulation such as cardiovascular disease, cancer, cerebral palsy (CP), and diabetes mellitus (DM) [45,46]. Limited research has been conducted on the relationship between exercise and enhancement of ANS cardiac regulation in pediatric clinical populations. A relatively recent study investigated the effects of an intense aerobic exercise trial consisting of bilateral upper and lower extremity activities on ANS cardiac regulation in children with unilateral CP [45]. Results indicated significant enhancement of the HRV PNS parameter, root mean square of successive differences (RMSSD), at post-intervention measurements which remained at the 3-month follow-up [45]. Additionally, there was improved walking endurance and upper extremity function of participants after completion of the trial. It was suggested that increased walking endurance may have not only increased cardiorespiratory fitness and ANS cardiac regulation but also enabled children to maintain exercise after the trial ceased, due to functional/physiological cardiorespiratory remodeling [45,47]. Furthermore, targeted upper extremity resistance training may have triggered increased activation of the exercise pressor reflex, allowing for vascular remodeling and cardiovascular adaptation [48,49]. Exercise also influenced ANS cardiac regulation in children with type 1 DM as demonstrated by Shin et al. Findings indicated that the 3-month walking program significantly increased total HRV (i.e., overall ANS activity) [50]. This is plausible considering the protective and/or therapeutic effects of exercise on hyperglycaemia [15]. Specifically, hyperglycemia damages cardiac autonomic neurons, blood vessels, and the heart via oxidative stress [15]. Notably, autonomic neuronal damage has been proposed to affect the vagus nerve first, leading to impaired PNS activity [16]. Increased exercise participation in children with type 1 DM may improve glycemia control through increased neuronal nitric oxide synthase [51,52]. This exercise-induced nitric oxide bioavailability may directly benefit ANS regulation via nerve lesion repair, reduced inflammation and improved vascular functioning and ANS regulation [15,53]. 

## 7. Recommendations for Exercise Prescription

General exercise guidelines recommend that children should be physically active for at least 60 min every day [20]. Whilst these guidelines are necessary and informative, there are currently no evidence-based exercise prescription recommendations for enhancing or optimizing child ANS cardiac regulation and overall health. The primary finding of the present review suggests that exercise-induced enhancement of ANS cardiac regulation (particularly PNS improvement) may be mediated by mechanisms related to neurogenesis [33,42]. These mechanisms appear to be stimulated by several cholinergic hormones and neurotransmitters, neurotrophins, and other metabolites at a neuronal/neurological, vascular and muscular level [15,24,33,40,41,49]. Findings demonstrate that enhancement of child ANS activity can occur in exercise programs ranging in frequency from 3 times/week to daily with session durations lasting from 20 min up to 2 h. Furthermore, improvement in ANS regulation can be seen in as little as 2 weeks of exercise training [39,45]. However, exercise training must be maintained to sustain improved ANS regulation [22,39]. These findings underlie the importance of manageability as a component for prescribing exercise to children and indicate that a feasible exercise program of relatively low volume (e.g., 3 times/week for approximately 30 min per session) is sufficient to enhance ANS cardiac regulation [14,27]. Furthermore, both aerobic and resistance exercise training benefit ANS regulation, although resistance training may have a greater influence on vagal enhancement whilst aerobically-based exercise may contribute more to sympathovagal balance [21,23,26,27,28,50]. Notably, exercise intensity may play a crucial role in modulating ANS regulation, particularly with respect to RMSSD. Given that most retrieved studies did not check for breathing frequency, exercise prescription based on RMSSD values would be most reliable [54]. The findings of this review demonstrated that there may be an optimal threshold for ANS enhancement, while exercise prescribed to children at maximal/supramaximal intensities may be inhibitory [17,18]. In addition, it is unknown whether low intensity exercise or combined (i.e., low, moderate and/or vigorous) exercise intensity prescription would have a favorable effect on ANS regulation. Taken together, moderate–vigorous exercise intensities should be prescribed to children to yield positive effects on ANS activity [14]. Lastly, population characteristics must be considered when prescribing exercise to children. For example, it may be unsafe to prescribe high volume, high intensity aerobic training to diabetic children due to metabolic and cardiovascular impairments and, as such, children with type 1 DM may enhance cardiac ANS regulation via low–moderate intensity exercise [53]. 

## 8. Conclusions

Expansive research has been conducted on exercise as a lifestyle modification for enhancing ANS cardiac regulation in adults but less is known about this in children. Moreover, current literature is limited regarding exercise prescription for enhancing child ANS cardiac regulation. Findings from this narrative review add to the current knowledge that exercise is an essential lifestyle modification component for improving ANS regulation and cardiometabolic health in children. Whilst volume (i.e., frequency and duration) is important, ANS enhancement via improving resting PNS regulation may largely depend on intensity and mode in accordance with population characteristics and goals. Additionally, sustained engagement in exercise during childhood is shown to be integral for maintaining enhanced ANS cardiac regulation and, as such, exercise prescription should reflect achievable programming. However, more research is required for further specification on the effects of exercise on ANS regulation, specifically research that focuses on different modes of training (e.g., resistance/strength, pilates, yoga, etc), low volume training (i.e., low intensity sessions, 1–2 times/week, 30-mins/session) and exercise in clinical populations. Furthermore, considering the limited number of studies investigating ANS reactivity to stress, measurement of HRV response to exercise may act as a valuable marker of child health that can be examined in future studies. Ideally, follow-up sessions to measure HRV after the intervention has ceased should also be incorporated to determine the lasting effects of the exercise prescription plan on ANS cardiac regulation.

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
