# Peer review of "Lifestyle Modification for Enhancing Autonomic Cardiac Regulation in Children: The Role of Exercise"

_children, 2019, doi:10.3390/children6110127_

Round 1

Reviewer 1 Report

This narrative review addresses mechanisms behind the effects of physical activity on cardiovascular disease, by examining the effects on ANS activity. The importance of increased knowledge on the this topic in children is well argued. I find the manuscript well structured, clear and easy to follow. I only have a few minor comments that may help clarify the understanding of some sentences.

Lines 63-64: The authors could add that this is a narrative review, to clearify this from the start.

Lines 90-92: I would move the sentence starting on line 91 ("Intensity of sessions..:") to the previous paragraph; this way, the first paragraph ends with descriptives of frequency (3 days/wk), duration (20min-2h), and then intensity (40-85%). Then, the next paragraph concerns the effects of these modes on ANS regulation. 

Line 219: Instead of "ANS cardiac regulation in children", I would prefer "ANS cardiac regulation in pediatric clinical populations" to specify that this paragraph deals with clinical populations.

Line 259: An N is missing in Notably.

Author Response

Reviewer 1: This narrative review addresses mechanisms behind the effects of physical activity on cardiovascular disease, by examining the effects on ANS activity. The importance of increased knowledge on the this topic in children is well argued. I find the manuscript well structured, clear and easy to follow. I only have a few minor comments that may help clarify the understanding of some sentences.

Point 1 - Lines 63-64: The authors could add that this is a narrative review, to clearify this from the start.

Response 1: Thank you for this comment. We agree that by adding in that this manuscript is a narrative review earlier (lines 63-64) will help the reader and, as such, have edited the manuscript to reflect this 

Point 2 - Lines 90-92: I would move the sentence starting on line 91 ("Intensity of sessions..:") to the previous paragraph; this way, the first paragraph ends with descriptives of frequency (3 days/wk), duration (20min-2h), and then intensity (40-85%). Then, the next paragraph concerns the effects of these modes on ANS regulation. 

Response 2: Thank you for this comment. We have moved the sentence "Intensity of sessions..." to the first paragraph of section 3 and agree that the first and second paragraph read  more cohesively now.

Point 3 - Line 219: Instead of "ANS cardiac regulation in children", I would prefer "ANS cardiac regulation in pediatric clinical populations" to specify that this paragraph deals with clinical populations.

Response 3: This is a great point and will clarify that this paragraph is specifically related to clinical populations. We have added in your suggestion on Line 222.

Point 4 - Line 259: An N is missing in Notably.

Response 4: Thank you for pointing this out. We have now corrected the spelling of Notably

Reviewer 2 Report

The current review article entitled “Lifestyle modification for enhancing autonomic cardiac regulation in children: The role of exercise” examined that which exercise modification for enhancing the cardiac function. This review by the author is a very important and interesting study. I think it is enough to attract the reader's attention through the current review, and the application of the exercise physiological point of view is positive. The somewhat unfortunate part is that I want the author to suggest data that can help readers check through tables and figures.

Author Response

Point 1: The current review article entitled “Lifestyle modification for enhancing autonomic cardiac regulation in children: The role of exercise” examined that which exercise modification for enhancing the cardiac function. This review by the author is a very important and interesting study. I think it is enough to attract the reader's attention through the current review, and the application of the exercise physiological point of view is positive. The somewhat unfortunate part is that I want the author to suggest data that can help readers check through tables and figures.

Response 1: Thank you very much for your comments and critique. We agree that a table of included articles would make it easier for readers to review the data. As such, we have created a summary table (see Table 1) outlining the author and date, participants, aim, exercise sub-topics and general findings. Additionally, we included a sentence in the methods section (lines 81-83) detailing the number of articles originally retrieved from the databases, the number of full texts analyzed and the final number of included studies.

Reviewer 3 Report

This is a systematic review of the literature regarding the role of exercise on ANS activity in children. Overall, it is well organized and follows a logical flow of information. English usage is good with minor errors.

General comments:

Please include the number of articles returned, analyzed, and accepted from the literature search into the methods section to give the reader perspective on the breadth of the review. Since information is organized into categories in the article, you might consider providing the number of articles returned/analyzed/reviewed for each category as well. The addition of a table summarizing and organizing the included articles would strengthen the article as well as provide a concise overview of reviewed literature to the reader. Including basic information on each article (author, date, sample size, topic/category, brief conclusion, etc.) should be a minimum. The first sentence of sections 5 (Intervention Duration) should be reworded or removed. As is, it is too vague and lacks direction. Consider adding "in children" to the end of the third sentence of sections 8 (Conclusion) (page 7, line 279). This is what you intended and what this review adds, yes?

Grammar:

A comma is needed on page 2, line 66 after "components" and before "this" The sentence on page 7, line 259 is lacking the first letter of "Notably". Add the "N".

Author Response

This is a systematic review of the literature regarding the role of exercise on ANS activity in children. Overall, it is well organized and follows a logical flow of information. English usage is good with minor errors.

General comments:

Point 1: Please include the number of articles returned, analyzed, and accepted from the literature search into the methods section to give the reader perspective on the breadth of the review. Since information is organized into categories in the article, you might consider providing the number of articles returned/analyzed/reviewed for each category as well. The addition of a table summarizing and organizing the included articles would strengthen the article as well as provide a concise overview of reviewed literature to the reader. Including basic information on each article (author, date, sample size, topic/category, brief conclusion, etc.) should be a minimum. The first sentence of sections 5 (Intervention Duration) should be reworded or removed. As is, it is too vague and lacks direction. Consider adding "in children" to the end of the third sentence of sections 8 (Conclusion) (page 7, line 279). This is what you intended and what this review adds, yes?

Response 1: Thank you for these comments and suggestions. We have now added in the number of articles originally retrieved from the databases, the number of full texts analyzed and the final number of included studies. We have also included a summary table (see Table 1) as you suggested including author and date, participants, aim, exercise sub-topics and general findings. We did consider dividing the results of the search strategy into exercise sub-topic categories, however, as you'll be able to see from the newly included table, many of the studies overlap in terms of their category so we thought it would be more clear to note which exercise sub-topics the articles addressed in the table. We agree that the first sentence of section 5 is vague and unnecessary. As such, we have deleted this sentence. We have also added "in children" to the end of the third sentence of section 8 as per your advice. 

Grammar:

Point 2: A comma is needed on page 2, line 66 after "components" and before "this" The sentence on page 7, line 259 is lacking the first letter of "Notably". Add the "N". 

Response 2: Thank you for bring this to our attention. We have added a comma after "components" on line 66 and corrected the spelling of "Notably".